# A note on the Wilcoxon-Mann-Whitney test and tied observations

**Markus Neuhäuser** [1,2] *, **Graeme D. Ruxton** [3]

**1** Department of Mathematics and Technology, RheinAhrCampus, Koblenz University of Applied Sciences, Remagen, Germany, **2** Institute for Medical Biometry, Informatics and Epidemiology, Medical Faculty, University of Bonn, Bonn, Germany, **3** School of Biology, University of St Andrews, St Andrews, United Kingdom

* neuhaeuser@rheinahrcampus.de

## Abstract

Recently, it was recommended to omit tied observations before applying the two-sample Wilcoxon-Mann-Whitney test McGee M. et al. (2018). Using a simulation study, we argue for exact tests using all the data (including tied values) as a preferable approach. Exact tests, with tied observations included guarantee the type I error rate with a better exploitation of the significance level and a larger power than the corresponding tests after the omission of tied observations. The omission of ties can produce a considerable change in the shape of the sample, and so can violate underlying test assumptions. Thus, on both theoretical and practical grounds, the recommendation to omit tied values cannot be supported, relative to analysing the whole data set in the same way whether or not ties occur, preferably with an exact permutation test.

## Introduction

Data points with identical values are called ties, they frequently occur. Recently, it was proposed to omit tied observations before applying the two-sample Wilcoxon-Mann-Whitney test [1]. Here, we argue against this recommendation.

Tied observations do not create insuperable problems for application of the Wilcoxon-Mann-Whitney test. It is true that the normal approximation of the Wilcoxon rank-sum can be poor when there are ties, although this depends on the number and pattern of ties [2]. However, with the advent of modern computing power, there is no need to use the normal approximation in order to apply this test. Indeed, even in the absence of ties, when sample sizes are small or moderate an exact permutation version of the test is preferable to the one that relies on asymptotic normality.

A permutation test is not only possible to calculate "the null distribution of the possible difference in means (or medians, or ratios, etc.)" [1], but also for obtaining the null distribution of the Wilcoxon rank-sum [3]. Neuhäuser [4] as well as Brunner at al. [2] explicitly demonstrate how the exact permutation test can be applied in the presence of ties. In the case of large sample sizes, an approximate permutation test can be applied based on a random sample of

code. Thus all simulations can easily be reproduced.

**Funding:** The author(s) received no specific funding for this work.

**Competing interests:** The authors have declared that no competing interests exist.

permutations (see e.g. [4]). Thus, there is no theoretical or practical need for omitting tied observations.

As noted by McGee [1] observations are often discretized or rounded, e.g. due to detection limits of the measurement instruments, even when the underlying distributions are continuous. In those cases, however, tied observations are not evenly distributed across samples (as in McGee's simulation study). In contrast, ties are more likely where the density of the distribution in higher. To omit tied values can thus produce a considerable change in the shape of distribution of the remaining data, compared to the whole data set. Therefore, solely because of this possible change of the distribution one should retain tied observations for the statistical analysis. This holds not only for the Wilcoxon-Mann-Whitney test, but also for other methods such as the t test. A fundamental assumption of statistical testing is that the sample is representative of the underlying population from which it is drawn. If ties are non-randomly distributed (as we argue they should generally be expected to be) then changing the sample by omitting tied values breaks this assumption, and the reduced sample is no longer representative of the underlying population that was originally of interest.

Moreover, as ties are non-randomly distributed in practice, the possible change in the shape of distribution of the remaining data after the omission of ties might lead to changes between groups in skewness, variance and other characteristics that can violate the shift alternative framework of the statistical testing approach.

In addition, when clinical trial data are analysed, the omission of tied observation would violate the intention-to-treat ideal of including all randomised subjects in the analysis. In this context, note that Mao [5] recently investigated the Wilcoxon-Mann-Whitney test in the presence of noncompliance. The noncompliers, included in an intention-to-treat analysis, are not an evenly distributed subset, and Mao [5] showed that the properties of the Wilcoxon-Mann-Whitney test depend on whether noncompliers are more likely in high-density regions or in the tails of the outcome distribution.

Furthermore, when rounding is carried out, the amount of rounding, i.e. the number of remaining decimal places, might be arbitrary. If rounding is necessary, we suggested that the "number of decimal places should be selected by the scientist on the basis of their understanding of the precision of measurements involved" [6, p. 298].

## Materials and methods

Like McGee [1], we performed a simulation study (using R). Again like McGee [1], we investigate the four distributions: normal (with variance 1), exponential (with rate 1), Cauchy, and Laplace (both with scale parameter 1). No changes in variability are considered, possible differences between the two groups are only location shifts. We investigate the Wilcoxon-Mann-Whitney test both based on complete samples with rounded values (and using mid-ranks), and after omission of tied observations. However, in contrast to McGee [1], tied observations are not evenly distributed across samples in our simulation, we round the simulated values to one or two decimal places, respectively, in order to simulate the creation of tied values by limited measurement precision. In contrast, in McGee's simulation a given number of values is randomly selected, regarded as ties and omitted (see e.g. the supplementary file SmallSample-PowerOmit.txt, available at https://github.com/MonnieMcGee/TiesInRankBasedTests/blob/master/R-Code/SmallSamplePowerOmit.txt). Thus, in McGee's simulations the percentage of tied observations was kept constant in the different scenarios. In our simulations, the number of ties varies, which we believe better reflects the situation in statistical practice.

We present results for balanced and unbalanced sample sizes, for the exact permutation test and the asymptotic test, as well as for two different significance levels, 0.01 and 0.05. All results

are based on 10000 simulation runs. Sometimes omission of tied observations caused one of the two groups to be empty. In this case no test could be performed. The estimated actual type I error rate or power, respectively, is estimated as the number of tests for which the p-value is less than the nominal level of significance divided by the number of performed tests. When rounding to two decimal places, empty groups did not occur. When rounding to one decimal place, empty groups occurred 0 to 207 times (with mean 21.4, and median 5) in 10000 simulations.

## Results and conclusion

The simulation results are displayed in Tables 1 to 4.

Although the tests after omission of tied observations do not have an inflated type I error rate, the tests with tied observations included have an actual size closer to the nominal significance level, especially when considering a significance level of 1%, and/or rounding to one decimal place. In general, the exact permutation test with tied observations included is a good choice in all the situations we investigate, the type I error is guaranteed, and the power is larger, often much larger, than the power of the corresponding tests after the omission of tied observations. With the advent of modern computing power, asymptotic tests provide no advantage to exact tests, but rely on assumptions about the data that will not always be met and often cannot be easily tested. In summary, the tests after the omission of tied observations show a lower exploitation of the significance level and a lower power. Hence, for realistic scenarios, they cannot be recommended with respect to type I error and power.

McGee [1] reported a more erratic, or worse, type I error rate and a reduced power when tied observations occur with a percentage 25% or 50% and are omitted. On the basis of those

**Table 1. Actual type I error and power for balanced sample sizes ($n_1 = n_2 = 10$), simulated observations rounded to 2 decimal places.**

| | Tied observations included | | | | Tied observations omitted | | | | |
| | exact | | asymptotic | | Average | exact | | asymptotic | |
| Shift | $\alpha = 5\%$ | $\alpha = 1\%$ | $\alpha = 5\%$ | $\alpha = 1\%$ | sample size | $\alpha = 5\%$ | $\alpha = 1\%$ | $\alpha = 5\%$ | $\alpha = 1\%$ |
|---|---|---|---|---|---|---|---|---|---|
| Normal distribution | | | | | | | | | |
| 0 | 0.045 | 0.009 | 0.050 | 0.006 | 18.9 | 0.042 | 0.009 | 0.050 | 0.007 |
| 0.5 | 0.17 | 0.05 | 0.19 | 0.05 | 19.0 | 0.16 | 0.05 | 0.18 | 0.04 |
| 1 | 0.52 | 0.27 | 0.55 | 0.24 | 19.1 | 0.50 | 0.25 | 0.53 | 0.23 |
| 1.5 | 0.86 | 0.65 | 0.88 | 0.60 | 19.2 | 0.84 | 0.62 | 0.86 | 0.59 |
| Exponential distribution | | | | | | | | | |
| 0 | 0.048 | 0.009 | 0.054 | 0.007 | 18.2 | 0.043 | 0.008 | 0.050 | 0.007 |
| 0.5 | 0.32 | 0.13 | 0.33 | 0.11 | 18.6 | 0.28 | 0.10 | 0.30 | 0.09 |
| 1 | 0.72 | 0.47 | 0.74 | 0.43 | 18.8 | 0.68 | 0.42 | 0.69 | 0.40 |
| 1.5 | 0.92 | 0.76 | 0.93 | 0.73 | 18.9 | 0.90 | 0.72 | 0.91 | 0.69 |
| Cauchy distribution | | | | | | | | | |
| 0 | 0.046 | 0.010 | 0.053 | 0.007 | 19.4 | 0.044 | 0.009 | 0.051 | 0.007 |
| 0.5 | 0.09 | 0.02 | 0.10 | 0.02 | 19.4 | 0.08 | 0.02 | 0.09 | 0.02 |
| 1 | 0.19 | 0.07 | 0.21 | 0.06 | 19.5 | 0.18 | 0.07 | 0.20 | 0.06 |
| 1.5 | 0.33 | 0.15 | 0.36 | 0.13 | 19.5 | 0.31 | 0.14 | 0.34 | 0.13 |
| Laplace distribution | | | | | | | | | |
| 0 | 0.045 | 0.011 | 0.052 | 0.009 | 19.1 | 0.043 | 0.011 | 0.051 | 0.009 |
| 0.5 | 0.13 | 0.04 | 0.14 | 0.03 | 19.1 | 0.13 | 0.04 | 0.14 | 0.03 |
| 1 | 0.39 | 0.18 | 0.41 | 0.15 | 19.2 | 0.36 | 0.17 | 0.39 | 0.15 |
| 1.5 | 0.68 | 0.42 | 0.70 | 0.38 | 19.3 | 0.65 | 0.39 | 0.68 | 0.37 |

**Table 2. Actual type I error and power for balanced sample sizes ($n_1 = n_2 = 10$), simulated observations rounded to 1 decimal place.**

| | Tied observations included | | | | | Tied observations omitted | | | |
| | exact | | asymptotic | | Average | exact | | asymptotic | |
| Shift | $\alpha = 5\%$ | $\alpha = 1\%$ | $\alpha = 5\%$ | $\alpha = 1\%$ | sample size | $\alpha = 5\%$ | $\alpha = 1\%$ | $\alpha = 5\%$ | $\alpha = 1\%$ |
|---|---|---|---|---|---|---|---|---|---|
| Normal distribution | | | | | | | | | |
| 0 | 0.047 | 0.009 | 0.048 | 0.007 | 11.9 | 0.039 | 0.007 | 0.047 | 0.004 |
| 0.5 | 0.18 | 0.05 | 0.18 | 0.05 | 12.1 | 0.13 | 0.03 | 0.14 | 0.03 |
| 1 | 0.54 | 0.28 | 0.54 | 0.25 | 12.6 | 0.38 | 0.15 | 0.41 | 0.13 |
| 1.5 | 0.87 | 0.65 | 0.87 | 0.62 | 13.3 | 0.69 | 0.39 | 0.72 | 0.36 |
| Exponential distribution | | | | | | | | | |
| 0 | 0.051 | 0.010 | 0.051 | 0.008 | 9.0 | 0.028 | 0.004 | 0.043 | 0.003 |
| 0.5 | 0.33 | 0.13 | 0.33 | 0.11 | 10.2 | 0.11 | 0.02 | 0.14 | 0.01 |
| 1 | 0.73 | 0.47 | 0.73 | 0.45 | 11.2 | 0.36 | 0.14 | 0.40 | 0.11 |
| 1.5 | 0.92 | 0.76 | 0.93 | 0.74 | 12.0 | 0.63 | 0.34 | 0.66 | 0.29 |
| Cauchy distribution | | | | | | | | | |
| 0 | 0.050 | 0.010 | 0.050 | 0.008 | 15.0 | 0.043 | 0.008 | 0.049 | 0.007 |
| 0.5 | 0.09 | 0.02 | 0.09 | 0.02 | 15.2 | 0.07 | 0.01 | 0.07 | 0.01 |
| 1 | 0.20 | 0.07 | 0.21 | 0.06 | 15.5 | 0.14 | 0.04 | 0.15 | 0.04 |
| 1.5 | 0.34 | 0.16 | 0.35 | 0.14 | 15.8 | 0.24 | 0.09 | 0.25 | 0.09 |
| Laplace distribution | | | | | | | | | |
| 0 | 0.050 | 0.011 | 0.051 | 0.010 | 12.8 | 0.041 | 0.008 | 0.048 | 0.007 |
| 0.5 | 0.14 | 0.04 | 0.14 | 0.04 | 13.0 | 0.09 | 0.02 | 0.10 | 0.02 |
| 1 | 0.40 | 0.18 | 0.40 | 0.16 | 13.5 | 0.26 | 0.09 | 0.27 | 0.08 |
| 1.5 | 0.70 | 0.42 | 0.70 | 0.39 | 14.0 | 0.49 | 0.24 | 0.51 | 0.22 |

**Table 3. Actual type I error and power for unbalanced sample sizes ($n_1 = 14$, $n_2 = 7$), simulated observations rounded to 2 decimal places.**

| | Tied observations included | | | | | Tied observations omitted | | | |
| | exact | | asymptotic | | Average | exact | | asymptotic | |
| Shift | $\alpha = 5\%$ | $\alpha = 1\%$ | $\alpha = 5\%$ | $\alpha = 1\%$ | sample size | $\alpha = 5\%$ | $\alpha = 1\%$ | $\alpha = 5\%$ | $\alpha = 1\%$ |
|---|---|---|---|---|---|---|---|---|---|
| Normal distribution | | | | | | | | | |
| 0 | 0.047 | 0.009 | 0.047 | 0.007 | 19.8 | 0.045 | 0.010 | 0.045 | 0.008 |
| 0.5 | 0.16 | 0.05 | 0.16 | 0.04 | 19.9 | 0.15 | 0.05 | 0.15 | 0.04 |
| 1 | 0.50 | 0.25 | 0.50 | 0.21 | 20.0 | 0.48 | 0.23 | 0.48 | 0.20 |
| 1.5 | 0.83 | 0.61 | 0.83 | 0.57 | 20.1 | 0.82 | 0.58 | 0.82 | 0.54 |
| Exponential distribution | | | | | | | | | |
| 0 | 0.048 | 0.010 | 0.048 | 0.008 | 19.0 | 0.046 | 0.010 | 0.046 | 0.008 |
| 0.5 | 0.28 | 0.10 | 0.28 | 0.08 | 19.4 | 0.24 | 0.08 | 0.24 | 0.06 |
| 1 | 0.71 | 0.42 | 0.71 | 0.38 | 19.6 | 0.66 | 0.36 | 0.66 | 0.32 |
| 1.5 | 0.93 | 0.76 | 0.93 | 0.72 | 19.7 | 0.91 | 0.70 | 0.91 | 0.65 |
| Cauchy distribution | | | | | | | | | |
| 0 | 0.044 | 0.010 | 0.044 | 0.008 | 20.3 | 0.043 | 0.011 | 0.043 | 0.009 |
| 0.5 | 0.08 | 0.02 | 0.08 | 0.02 | 20.4 | 0.08 | 0.02 | 0.08 | 0.02 |
| 1 | 0.18 | 0.06 | 0.18 | 0.05 | 20.4 | 0.17 | 0.06 | 0.17 | 0.05 |
| 1.5 | 0.32 | 0.14 | 0.32 | 0.12 | 20.5 | 0.30 | 0.13 | 0.30 | 0.11 |
| Laplace distribution | | | | | | | | | |
| 0 | 0.045 | 0.009 | 0.045 | 0.007 | 20.0 | 0.043 | 0.008 | 0.044 | 0.007 |
| 0.5 | 0.14 | 0.04 | 0.14 | 0.03 | 20.0 | 0.13 | 0.04 | 0.13 | 0.03 |
| 1 | 0.39 | 0.18 | 0.39 | 0.15 | 20.1 | 0.37 | 0.17 | 0.37 | 0.15 |
| 1.5 | 0.66 | 0.41 | 0.66 | 0.37 | 20.2 | 0.64 | 0.39 | 0.64 | 0.35 |

**Table 4. Actual type I error and power for unbalanced sample sizes ($n_1 = 14$, $n_2 = 7$), simulated observations rounded to 1 decimal place.**

| | Tied observations included | | | | | Tied observations omitted | | | |
|---|---|---|---|---|---|---|---|---|---|
| | exact | | asymptotic | | Average | exact | | asymptotic | |
| Shift | $\alpha$ = 5% | $\alpha$ = 1% | $\alpha$ = 5% | $\alpha$ = 1% | sample size | $\alpha$ = 5% | $\alpha$ = 1% | $\alpha$ = 5% | $\alpha$ = 1% |
| Normal distribution | | | | | | | | | |
| 0 | 0.047 | 0.009 | 0.047 | 0.008 | 12.2 | 0.039 | 0.006 | 0.045 | 0.004 |
| 0.5 | 0.16 | 0.05 | 0.16 | 0.04 | 12.3 | 0.11 | 0.02 | 0.12 | 0.02 |
| 1 | 0.51 | 0.24 | 0.51 | 0.22 | 12.8 | 0.35 | 0.12 | 0.37 | 0.09 |
| 1.5 | 0.84 | 0.60 | 0.84 | 0.58 | 13.5 | 0.66 | 0.35 | 0.68 | 0.30 |
| Exponential distribution | | | | | | | | | |
| 0 | 0.046 | 0.011 | 0.046 | 0.009 | 9.2 | 0.024 | 0.002 | 0.038 | 0.002 |
| 0.5 | 0.28 | 0.10 | 0.28 | 0.08 | 10.1 | 0.05 | 0.01 | 0.07 | 0.005 |
| 1 | 0.72 | 0.42 | 0.72 | 0.39 | 11.1 | 0.22 | 0.05 | 0.25 | 0.04 |
| 1.5 | 0.94 | 0.76 | 0.94 | 0.73 | 11.8 | 0.49 | 0.20 | 0.54 | 0.17 |
| Cauchy distribution | | | | | | | | | |
| 0 | 0.045 | 0.010 | 0.045 | 0.009 | 15.6 | 0.040 | 0.008 | 0.043 | 0.006 |
| 0.5 | 0.08 | 0.02 | 0.08 | 0.02 | 15.7 | 0.06 | 0.01 | 0.06 | 0.01 |
| 1 | 0.18 | 0.06 | 0.18 | 0.05 | 16.0 | 0.12 | 0.04 | 0.12 | 0.03 |
| 1.5 | 0.32 | 0.14 | 0.32 | 0.12 | 16.3 | 0.22 | 0.08 | 0.22 | 0.07 |
| Laplace distribution | | | | | | | | | |
| 0 | 0.046 | 0.009 | 0.046 | 0.007 | 13.2 | 0.038 | 0.006 | 0.042 | 0.003 |
| 0.5 | 0.14 | 0.04 | 0.14 | 0.03 | 13.4 | 0.09 | 0.02 | 0.10 | 0.02 |
| 1 | 0.39 | 0.17 | 0.39 | 0.16 | 13.8 | 0.26 | 0.08 | 0.27 | 0.07 |
| 1.5 | 0.66 | 0.40 | 0.66 | 0.38 | 14.3 | 0.48 | 0.22 | 0.49 | 0.19 |

results, one might advocate omitting ties only when the percentage of ties in the sample is less than 15%. However, such a strategy would often preclude defining the statistical approach ahead of data collection, since the percentage of ties would normally be difficult to predict with any confidence. Further, the exact test with tied observations included has preferable type I error and power values than the test performed after omission of ties in the scenarios presented in Tables 1 and 3 where the average proportion of ties is not larger than 10%. Thus, the exact permutation test is always a good choice. This test is available in various statistical software packages including the open-source statistical software R. In addition, a free and easy-to-use online calculator that provides the exact permutation Wilcoxon-Mann-Whitney test is available at https://ccb-compute2.cs.uni-saarland.de/wtest/ [7].

When applying this permutation test there is, even in the case of discrete or ordinal data, no need, and no reason, to replace the test statistic. Thus, the permutation test should be performed with the Wilcoxon rank sum, not with the difference in means, medians, or ratios. The exact permutation Wilcoxon-Mann-Whitney test can be applied even in the extreme case of binary data: when two groups are compared based on binary data the standard method is Fisher's exact test; this test can be considered as a special case of the exact version of the Wilcoxon-Mann-Whitney test; both tests result in identical p-values [2, p. 106].

It should be noted that McGee [1] also investigated other methods of handling ties. Her detailed study shows some advantages and disadvantages of the various procedures. McGee [1] concluded that, aside from omission of tied observations, the next best option would be jittering the data, i.e. adding random noise to the observations to break ties. However, an exact permutation test with mid-ranks is preferable to using randomly broken ties [4, 6, 8].

To conclude, we see no reason to recommend the omission of tied observations when applying the Wilcoxon-Mann-Whitney test, and good reason not to recommend such omission, even when the number of ties is low. Exact testing of the whole dataset regardless of ties is a better option.

## Supporting information

**S1 File. R code for simulation.**
(PDF)

## Author Contributions

**Conceptualization:** Markus Neuhäuser.

**Formal analysis:** Markus Neuhäuser.

**Investigation:** Markus Neuhäuser, Graeme D. Ruxton.

**Methodology:** Markus Neuhäuser, Graeme D. Ruxton.

**Software:** Markus Neuhäuser.

**Validation:** Graeme D. Ruxton.

**Writing – original draft:** Markus Neuhäuser.

**Writing – review & editing:** Graeme D. Ruxton.

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
