## [Decision Letter · Decision Letter 0]

18 Mar 2024

PONE-D-23-33167A note on the Wilcoxon-Mann-Whitney test and tied observationsPLOS ONE

Dear Dr. Neuhäuser,

Thank you for submitting your manuscript to PLOS ONE. After careful consideration, we feel that it has merit but does not fully meet PLOS ONE’s publication criteria as it currently stands. Therefore, we invite you to submit a revised version of the manuscript that addresses the points raised during the review process. See specific comments below.

We look forward to receiving your revised manuscript.

Kind regards,

Ben Ridenhour

Academic Editor

PLOS ONE

Journal Requirements:

Additional Editor Comments:

Both reviewers have very minor comments that I would prefer that you address prior to publication. However, all changes are minor and both reviewers are positive about publishing this note; please address their comments and resubmit for acceptance.

Reviewers' comments:

Reviewer's Responses to Questions

**Comments to the Author**

1. Is the manuscript technically sound, and do the data support the conclusions?

Reviewer #1: Yes

Reviewer #2: Yes

2. Has the statistical analysis been performed appropriately and rigorously? 

Reviewer #1: Yes

Reviewer #2: Yes

3. Have the authors made all data underlying the findings in their manuscript fully available?

Reviewer #1: No

Reviewer #2: Yes

4. Is the manuscript presented in an intelligible fashion and written in standard English?

Reviewer #1: Yes

Reviewer #2: Yes

5. Review Comments to the Author

Reviewer #1: This short note concerns the use of the Wilcoxon-Mann-Whitney (WMW) test in the presence of ties. Previously, McGee (2018) recommended removing the ties from analysis on the grounds that they interfere with the normal approximation. In this paper, however, the authors use a simulation to show that removal of ties generally leads to undesirable properties of the test. It is concisely and well written, and I am in favor of publishing it.

As a side note, the idea of keeping all data in the analysis rather than focusing on a subset (in this case untied observations) is captured in the famous intent-to-treat (ITT) principle. Recently, Mao (2022) studied the properties of the WMW test under the ITT principle in the presence of noncompliance. Although this is a slightly different scenario, the authors may consider connecting to this literature to strengthen their case and bring it up to date.

Mao, L. (2022). On the relative efficiency of intent-to-treat Wilcoxon–Mann–Whitney test in the presence of non-compliance. Biometrika, 109, 873-880.

Minor:

"In is true that" -> "It is true that"

Reviewer #2: Overall this is a clear and concise paper that makes a very clear point about not omitting ties when using the Wilcoxon-Mann-Whitney test. The limitations and flaws of the conclusions from McGee 2018 are clear. The only suggestion I would have pertains to something that is not clear to me about McGee's methodology, even after having read their paper. It is not clear how McGee achieved uniformly distributed ties, even after having read the paper. Perhaps this is more clear from the code, which I started to examine, but I assume the authors of this paper have had a closer look. The authors make a point of questioning whether McGee's simulation study considered realistic scenarios. I think maybe a little more needs to be said about those (un)realistic scenarios.

6. PLOS authors have the option to publish the peer review history of their article (what does this mean?). If published, this will include your full peer review and any attached files.

Reviewer #1: No

Reviewer #2: No

---

## [Author Response · Author response to Decision Letter 0]

19 Apr 2024

As requested, we uploaded a separate file labeled "Response to Reviewers" with our response.

---

## [Decision Letter · Decision Letter 1]

12 Jun 2024

PONE-D-23-33167R1A note on the Wilcoxon-Mann-Whitney test and tied observationsPLOS ONE

Dear Dr. Neuhäuser,

Thank you for submitting your manuscript to PLOS ONE. After careful consideration, we feel that it has merit but does not fully meet PLOS ONE’s publication criteria as it currently stands. Therefore, we invite you to submit a revised version of the manuscript that addresses the points raised during the review process. See below for my comments and those of the reviewers.

We look forward to receiving your revised manuscript.

Kind regards,

Benjamin Jerry Ridenhour

Academic Editor

PLOS ONE

Journal Requirements:

Additional Editor Comments:

Please address the useful comments made by the reviewers. Please note that the third reviewer is the author of the original article; while I think she brings up some valid concerns, it seems as if it comes down to some qualifiers/caveats as to when exclusion of ties might be useful. Furthermore, she makes that point that in the original article other methods of dealing with ties was explored (and not simply removal). Improved discussion of these caveats and other contributions of the original work would better couch/place the criticism of the current manuscript within the previous work.

Reviewers' comments:

Reviewer's Responses to Questions

**Comments to the Author**

1. If the authors have adequately addressed your comments raised in a previous round of review and you feel that this manuscript is now acceptable for publication, you may indicate that here to bypass the “Comments to the Author” section, enter your conflict of interest statement in the “Confidential to Editor” section, and submit your "Accept" recommendation.

Reviewer #3: (No Response)

2. Is the manuscript technically sound, and do the data support the conclusions?

Reviewer #3: Partly

3. Has the statistical analysis been performed appropriately and rigorously? 

Reviewer #3: No

4. Have the authors made all data underlying the findings in their manuscript fully available?

Reviewer #3: Yes

5. Is the manuscript presented in an intelligible fashion and written in standard English?

Reviewer #3: Yes

6. Review Comments to the Author

Reviewer #3: I have given a detailed response in an attached PDF. In brief, I believe that the author's results are different from McGee 2018 partially because their percentage of tied observations varies widely between simulations. Also, I agree that permutation tests are a viable option when there are large numbers of ties between or within samples.

7. PLOS authors have the option to publish the peer review history of their article (what does this mean?). If published, this will include your full peer review and any attached files.

Reviewer #3: **Yes: **Monnie McGee

---

## [Author Response · Author response to Decision Letter 1]

18 Jul 2024

Our detailed response to the reviewer can be found in a separate document.

In response to your e-mail sent on July 9, the references are now placed at the end of the manuscript (last page, i.e. page 12) and are numbered in the order that they appear in the text. 

The tables are part of the main manuscript, see pages 8-11. In response to your e-mail sent on July 9 we included a copy of Tables 1,2,3 and 4 as a separate file. Then, in your e-mail sent on July 11 you wrote "please include your tables as part of your main manuscript and remove the individual files". Therefore, we removed the individual file with the tables, however, all tables are still part of the main manuscript. Now, in your e-mail sent on July 12, you again asked to "include a copy of Tables 1,2,3 and 4". In the same e-mail you asked to "include your tables as part of your main manuscript and remove the individual files."

Therefore, now all four tables are part of the main manuscript. We do not have supplementary tables and therefore no separate files.

---

## [Editor Report · Decision Letter 2]

6 Aug 2024

A note on the Wilcoxon-Mann-Whitney test and tied observations

PONE-D-23-33167R2

Dear Dr. Neuhäuser,

We’re pleased to inform you that your manuscript has been judged scientifically suitable for publication and will be formally accepted for publication once it meets all outstanding technical requirements.

Kind regards,

Benjamin Jerry Ridenhour

Academic Editor

PLOS ONE
---

## [Editor Report · Acceptance letter]

12 Aug 2024

PONE-D-23-33167R2 

PLOS ONE

Dear Dr. Neuhäuser, 

I'm pleased to inform you that your manuscript has been deemed suitable for publication in PLOS ONE. Congratulations! Your manuscript is now being handed over to our production team.

Kind regards, 

on behalf of

Dr. Benjamin Jerry Ridenhour 

Academic Editor

PLOS ONE